# Everything, everywhere, all at once - Surveillance and molecular epidemiology reveal *Melissococcus plutonius* is endemic among Michigan, US beekeeping operations

Peter D. Fowler[1]*, Upasana Dhakal[2‡], Jeff H. Chang[2‡], Meghan O. Milbrath[1,3]

**1** Comparative Medicine and Integrative Biology, College of Veterinary Medicine, Michigan State University, East Lansing, Michigan, United States of America, **2** Department of Botany and Plant Pathology, Oregon State University, Corvallis, Oregon, United States of America, **3** Department of Entomology, Michigan State University, East Lansing, Michigan, United States of America

☯ These authors contributed equally to this work.
‡ UD and JHC, MOM also contributed equally to this work.
* fowlerpe@msu.edu

## Abstract

European foulbrood (EFB) is a severe bacterial disease of honey bee brood often leading to significant declines in colony health and honey production. The dearth of data on this disease in the United States (US) complicates response efforts. In 2021 and 2022 we conducted a two-year cross-sectional surveillance study among Michigan beekeepers to establish baseline pathogen and disease prevalence. We combined this surveillance with molecular epidemiology to investigate genetic diversity, and transmission dynamics of *Melissococcus plutonius*, the causative agent of EFB, in US honey bee colonies. PCR screening detected *M. plutonius* in all 15 migratory and stationary beekeeping operations and in 6 of 14 hobby beekeeping operations. Infection and disease were found to be seasonal, with prevalence of both peaking in June when over half the colonies were infected, and over 20% had clinical EFB. Whole genome, single nucleotide polymorphism analysis revealed wide genetic diversity even within a single hive. Operations often had multiple genotypes present which varied from year to year, consistent with high rates of transmission and reinfection. Prevalence and whole genome data provided here will be critical in tracking the efficacy of mitigation efforts and underscore the necessity of additional epidemiological investigations.

## 1. Introduction

Honey bees (*Apis mellifera*) are vital for crop pollination and ecosystem services worldwide [1]. Over the last two decades beekeepers in the United States (US) have been experiencing high rates of colony losses [2] that threaten sustainability and

**Data availability statement:** All supporting data can be found in supplementary data files. Genomic data generated was deposited under BioProject PRJNA1195964 : Honey bee pathogen: Melissococcus plutonius assemblies - United States 2019-2023. The development version of the pipeline used in our analysis pathogensurveillance can be accessed at: https://nf-co.re/pathogensurveillance/dev/.

**Funding:** The work done by Peter Fowler and Meghan Milbrath was funded through The Rackham Research Endowment Program, Project GREEEN (Project GREEEN Proposal #GR21-033) and North Central SARE (Award #GNC21-325). Work in the Chang lab performed by Upasana Dhakal and Jeff H. Chang was supported by a grant to J. H. C. from the National Institutes of Food and Agriculture, US Department of Agriculture (Award 2023-51181-41170). The funder had no role in manuscript preparation or the decision to publish.

**Competing interests:** The authors have declared that no competing interests exist.

national food security. One driver of this loss is European foulbrood (EFB) [3], a disease of honey bee brood (larvae and pupae) caused by the Gram-positive bacterium *Melissococcus plutonius* (Ex. White). Honey bee colonies afflicted with EFB have a significant reduction in growth over the honey production season [4], resulting in unsustainable economic losses for beekeepers [5]. Complicating this, some colonies with viable *M. plutonius* present never go on to develop disease [4], suggesting that infection may be more widespread than previously thought based on clinical symptoms alone.

EFB has plagued beekeepers in North America for over a century [6], yet little is known about prevalence or diversity of *M. plutonius* circulating in the United States and its impact on clinical disease. In a 2018 study in Chihuahua, Mexico, de León-Door et al. reported *M. plutonius* infection in 34% of the 154 bee colonies screened, suggesting infection rates may be higher than expected in North America [7]. Recent passive surveillance data identified EFB in 44 US states [8], finding highly variable state-level infection rates among submitted samples, ranging from 0% to 63%. Active disease surveillance in Oregon, US blueberry fields also found 41–53% of bee colonies with clinical EFB [9]. Recent research in Michigan found similarly high levels of EFB around June [4], but this was restricted to a single operation and the prevalence among Michigan and US beekeeping operations more broadly remains unknown.

*Melissococcus plutonius* is a phenotypically and genotypically diverse species, broadly classified as two unique strain (genetic variant) groupings, typical and atypical, with atypical strains capable of growth in aerobic conditions [10]. The virulence of multiple strains has been studied in vitro and found to vary widely with atypical strains maintaining high levels of virulence, while typical strains vary from asymptomatic to highly virulent [11–17]. The pMP19 plasmid has been implicated in virulence for some typical strains [16], but atypical virulence mechanisms remains unknown [18]. *M. plutonius* has also been classified into sequence types based on multi-locus (four genes) sequence typing (MLST) scheme [19]. Sequence types (ST) are further subdivided into genetically clustered clonal complexes (CC), with atypical strains containing a single clonal complex (CC12) and typical strains containing two (CC3 and CC13) [20]. To date over 40 unique sequence types have been reported across the globe [12] however only seven isolates from the US have been included in this analysis, representing only three strains: atypical ST12 and ST10 and typical ST3 [19]. To better understand the baseline prevalence of EFB in regional US beekeeping operations (objective 1), we undertook a cross-sectional surveillance study involving beekeeping operations based in the US state of Michigan from 2021–2022. Our second objective was to determine if EFB infection and subsequent transmission risk were present in honey bee colonies during interstate travel. To fulfill this objective, we surveyed commercial honey bee colonies from around the US while in California for almond pollination (February), and Michigan based operations following almond pollination after travel to Georgia (April) prior to returning to Michigan. Finally, a subset of *M. plutonius* isolates from the current surveillance study, and isolates collected from previous projects dating back to 2019 underwent whole genome analysis to both infer phylogenetic relationships, and to provide insights into pathogen diversity,

transmission, and persistence (objective 3). The baseline prevalence data and population structure of *M. plutonius* provided in this study are valuable resources for studies on the epidemiology and mechanisms of virulence, offering new avenues for research into disease mitigation.

## 2. Results

### 2.1 Prevalence of *Melissococcus plutonius* in Michigan (Objective 1)

*Melissococcus plutonius* is highly pervasive in Michigan honey bee colonies; 36% (n = 308) of the 855 honey bee colonies screened as part of the 2021 and 2022 cross-sectional surveillance in Michigan, tested positive for *M. plutonius* by duplex PCR (Fig 1). However, viable *M. plutonius* was recovered from only 56% (n = 171) of these hives. Overall infection rates were 9% higher in 2022 (40%) than in 2021 (31%) ($p < 0.01$, chi-squared = 7.899, df = 1). *Melissococcus plutonius* was detected in every beekeeping operation sampled over the two-year period, but one operation (J) only had 3 positive colonies in 2021 and no positive colonies in 2022. Five beekeepers (A, C, F, K and N) had *M. plutonius* detected in every yard sampled both years, however rates of infection varied widely by yard. For example, operation C had one yard with 94% (17/18) of sampled colonies infected and another yard with 10% (1/10) of sampled colonies infected. Of 16 hives sampled from 11 hobby operations, *M. plutonius* was identified in 7 hives from five beekeepers (7/16). Total hives included in surveillance is summarized in S1 Table.

Of the 855 colonies screened from 2021 to 2022, 787 were inspected for clinical disease, 284 of these screened positive for *M. plutonius* by duplex PCR, and 28% of these positive colonies (n = 80) had signs of EFB at the time of sampling. The remaining 72% (n = 204) of infected colonies had no symptoms of EFB, suggesting *M. plutonius* is highly pervasive even among asymptomatic colonies. Mean infection rate among asymptomatic colonies in yards where active clinical EFB disease was present was greater (30.6%) than in yards with no clinical EFB (19.4%) ($p < 0.001$, chi-square = 82.167, df = 1) (Fig 2). The two operations with the highest number of infected colonies (A and C) also had the highest rates of clinical disease. For operation A, all but one yard in the study had sampled colonies with clinical disease. For operation C, no

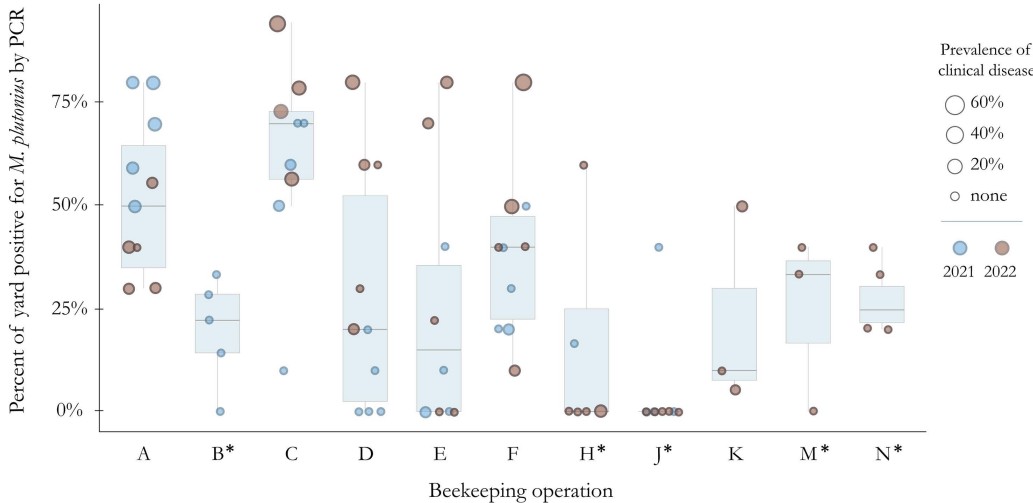

**Fig 1. Prevalence of *M. plutonius* varies widely by yard and operation for migratory (A, C, D, E, F, K) and stationary (B, H, J, M, N, marked with*) beekeepers in Michigan (2021-2022).** Each dot represents a yard grouped by operation (x-axis) with the percentage of colonies in the yard positive for *M. plutonius* by duplex PCR (y-axis). Boxplots show distribution and median values for the plotted yards. Operations with fewer than 3 yards (G, I, X) and yards with fewer than 5 colonies (L) were removed from this figure. Operation N had no information on clinical disease; size is not representative of disease prevalence for this operation.

clinical disease was observed in three yards. Two of these yards had very high rates of infection, with 7 of the 10 colonies screening positive for *M. plutonius,* suggesting high rates of asymptomatic infection. Operations B, J, M and N had no colonies with signs of EFB despite having some yards with over 25% of colonies infected. Seven colonies of the 855 were recorded as having signs consistent with larval disease but screened negative by duplex PCR for EFB. Surveillance carried out here was mostly exploratory in nature and was intended to get a broad view of the scope of the problem of EFB in Michigan. For that reason, comparisons between operation types are not likely to provide meaningful insights.

### 2.2 Prevalence of *Melissococcus plutonius* during almond pollination (Objective 1)

In February of 2023, colonies were sampled and inspected during almond pollination in California to provide insights into transmission potential during this period. Of the 144 colonies sampled from 22 operations originating from 8 different states, 11% (n = 13) screened positive for *M. plutonius* (S2 Table). Rates differed widely but one site from an Oregon-based operation had brood disease evident in all hives sampled, with *M. plutonius* being detected in four of the five. Another site from a separate Oregon based operation had *M. plutonius* present in three of the five colonies sampled, two of which had clinical disease.

### 2.3 Seasonality of EFB infection and disease (Objective 2)

Data from all years and multiple studies were pooled to identify seasonal fluctuations in pathogen and disease prevalence. Both *M. plutonius* infection and clinical EFB disease showed strong seasonality, peaking in June (Fig 3). Infection and disease were found in all months that sampling occurred: as early as February, and as late as August. April and May data were from only two operations: April data include 128 colonies screened in Georgia in 2022 from two Michigan-based operations: 99 colonies from operation C and 29 from operation K. Of the 99 colonies from operation C, 24% (n = 24) were positive by PCR while 9% (n = 9) had clinical EFB. Of the 29 from operation K, 35% (n = 10) were positive by PCR while 10% (n = 3) had clinical EFB. In May, we found increased prevalence of both pathogen (44%) and clinical disease (14%) in colonies from operation C after arriving in Michigan. June through August include data from operation C as well

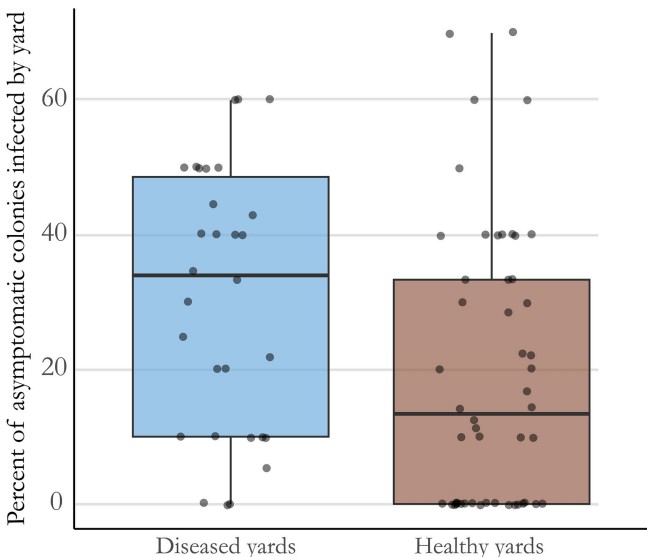

**Fig 2. Percentage of asymptomatic colonies sampled in each yard that were infected as determined by PCR detection of *Melissococcus plutonius*, grouped by disease status of the yard.** Each dot represents a yard where 5 or more colonies were sampled. Yards with at least one symptomatic colony (more than 10 larvae affected) are on the left (blue), while yards where no clinical EFB was found are on the right (brown).

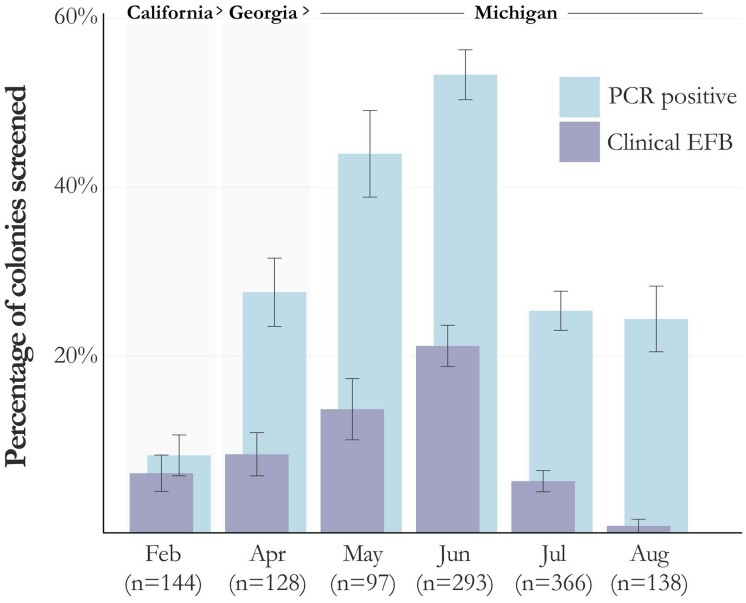

**Fig 3. Rates of both *Melissococcus plutonius* and clinical EFB among sampled colonies peaks in June.** Bars represent the percent of colonies inspected in each month that screened positive by duplex PCR for *M. plutonius* (light blue – right) and the percentage that had clinical EFB at the time of sampling (purple – left). Error bars represent the standard error for each month.

as additional beekeeping operations from around Michigan. In June, overall rates increased to 54% pathogen prevalence and 22% disease prevalence before both rates dropped significantly in July (26% pathogen, 6% disease) and August (26% pathogen, < 1% disease).

## 2.4  Impact of strain type on clinical disease (Objective 3)

We identified strain type (typical vs atypical) of the 284 PCR positive bee colonies that had associated health data. Of these, 48% (n = 137) tested positive for typical only, 13% (n = 36) tested positive for atypical only, and 39% (n = 111) had both typical and atypical strains detected (mixed infection) (Fig 4). We saw similar distributions in each region sampled: California – 9 typical, 2 atypical, and 2 mixed infections; Georgia – 19 typical and 17 mixed infections, and no atypical only

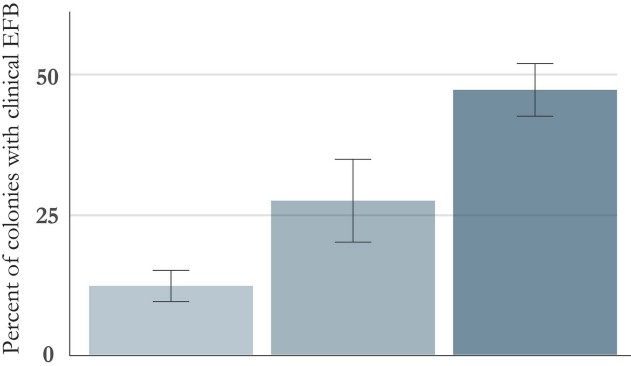

**Fig 4. Proportion of colonies with clinical disease by duplex PCR results.** 'Both' represents colonies that screened positive for both typical and atypical strains. Error bars represent standard error.

infections. The highest disease rate was in colonies co-infected with both strain types (48%) followed by colonies with only atypical (28%), and the lowest rate was among colonies that only tested positive for typical strains (12%) (p < 0.001, chi-squared 37.849, df = 2). Odds ratio for diseased colonies with atypical only infections was 2.70 (95% CI: 0.99–7.11) and 6.40 (95% CI: 3.31–12.88) for colonies with mixed infections when compared with those with typical only infections.

## 2.5 Population genetics of *Melissococcus plutonius* (Objective 3)

For a higher-resolution inference on phylogenetic relationships and insights into pathogen diversity, transmission, and persistence, we generated whole genome sequences for a subset of isolates. A total of 96 isolates was selected, including those from hives sampled in this study (2022–2023), as well as those from previous sampling efforts in 2019 [21] and 2020 [4]. Selected isolates represent the broadest temporal and geographic range, with one to two isolates from each operation, across multiple yards and multiple years. Initial use of read *k*-mers to infer taxonomic identities suggested that 94 of the 96 sequenced strains are members of the *M. plutonius* species. These 94 each assembled into 31 to 114 contigs (average of 51 contigs), mean N50 was 323 kb. Genomes averaged 2.1 Mb in size with an average of 1995 predicted genes (coding and non-coding). Two samples were heavily contaminated and three did not assemble well and were dropped from further analyses.

## 2.6 Core genome analysis

Next, we generated a core genome phylogeny to infer relationships among collected isolates and with typed isolates that have publicly available genome sequences. The typical and atypical strains separated into two distinct clades in a core genome maximum likelihood tree (Fig 5). Atypical strains collected in this study grouped with publicly available strains previously identified as ST12 (C12), ST19 (CC12) while the typical strains grouped with those typed as ST3 (CC3), ST39 (CC3).

## 2.7 Whole genome SNP analysis

Next, we mapped collection variables onto the minimum spanning networks and searched for patterns that could yield insights into disease spread (Fig 6). Most operations did not appear to be associated with specific genotypes and in contrast were associated with remarkable diversity of *M. plutonius.* Of the 19 operations that were sampled, 15 were associated with two or more sequenced strains and of these, 11 had multiple genotypes. Four of these fifteen operations with multiple strains had both typical and atypical strains, including operation G where only two isolates were sequenced. We sequenced 18 strains from operation C, and they were classified into four typical and seven atypical genotypes. Two operations had genotypes unique to that operation.

Within the three operations where sample collection spanned over three or four years (A, C, and F), we found diverse genotypes with little consistency from year to year. For example, operation F was sampled over four consecutive years and showed large variation in the genotype of strains isolated year after year. Only strains of atypical genotypes 14 and 33 were isolated in more than one year. Similarly, from operation A, we identified only one atypical genotype, 26, that had members isolated in separate years. There was no common genotype isolated from operation C among the years. However, given the diversity within operations, the differences in genotype composition across years could be reflective of limited sampling.

There were some epidemiological links among operations. Strains of seven atypical genotypes (26, 35, 44, 46, 53, 57, and 60), were isolated from more than one operation (Fig 6). Genotype 26 (atypical) was the most frequently sampled and included 13 isolates from 11 colonies collected from operations A, C, and D in 2022 and 2023. One isolate is from a colony sampled in California during almond pollination, two were sampled from operation C while in a holding yard in Georgia and the remaining isolates were from honey bee colonies sampled in Michigan. Colonies which screened positive in California during almond pollination originated from 6 different beekeeping operations: three from Michigan, two from Oregon, and one from North Dakota. Whole genome SNP analysis revealed genetic distance based on state of origin, which may indicate differences at larger spatial scales.

**Fig 5. Core gene phylogenetic tree of *Melissococcous plutonius* strains (sequenced for this study and publicly available).** * Sequence type was determined by MLST and labeled along the tips of the tree. Heatmap shows if the strain is typical or atypical; presence, absence and subtype of pMP1 and pMP19 plasmids.

*M. plutonius* strains can also be diverse within a single honey bee colony. Agar culture from a single larval sample from one honey bee colony revealed two separate colony morphologies with unique growth characteristics. Whole genome SNP analysis comparing the two isolates revealed two genetically distinct genotypes, a faster growing genotype 70, and a comparatively slower growing genotype 13 (genotypes marked by *, Fig 6). Both genotypes fall under atypical MLST ST19 and are 109 SNPs apart.

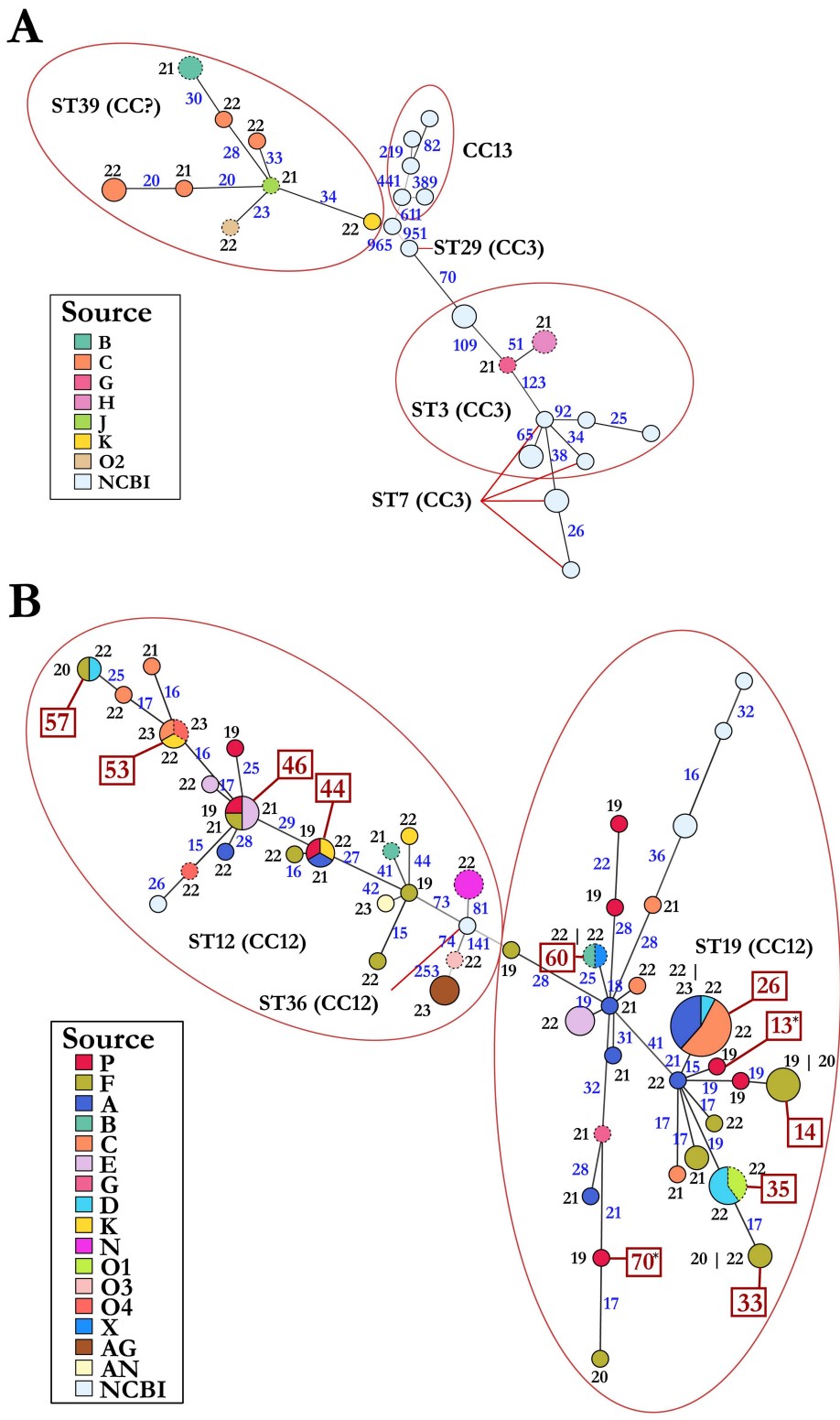

**Fig 6. *Melissococcus plutonius* genotypes are distributed broadly across the US. Minimum spanning networks for (A) typical isolates and (B) atypical isolates.** Nodes represents genotypes and are scaled according to the number of genotypes. Nodes are colored proportionally to the source of

the genotypes (see keys). Nodes outlined in solid lines represent isolates from migratory operations. Nodes outlined in dashed lines represent isolates from stationary operations. The numbers in black and along nodes indicate the years of isolation 20xx. *: two different genotypes isolated from a single larval sample. The numbers in blue and along edges are the numbers of SNPs separating two genotypes.

Isolates identified as ST39 group into genotypes that are more distant than the distances between other clonal complexes of typical strains (Fig 6). Prior MLST analyses describe members of ST39 as members of both CC3 and CC13 [13,21]. We suggest that ST39 represents a novel clonal complex. Typical genotypes of ST3 and ST7 are closely related and do not separate clearly. Interestingly, seventeen of the atypical isolates were cultured from larval samples that screened positive for typical strains only suggesting the duplex PCR is not sufficiently sensitive to detect atypical *M. plutonius* in the presence of typical strains.

### 2.8 Plasmid analysis

Three different plasmids (pMP1, pMP19, pMP43) have been identified in *M. plutonius* [11,22]. pMP1 (~200 kb) is present in all previously sequenced strains while pMP19 (~20 kb) is variable in presence. We identified contigs corresponding to pMP1 and pMP19 plasmids and findings were consistent with previous research [22] in confirming that pMP1 plasmid is present in all analyzed strains. Next, we used a k-mer analysis that reflects variations in sequence and gene content to classify pMP1 into subtypes. This plasmid separated into two subtypes and the separation correlated with the division of *M. plutonius* into atypical and typical groups (Fig 7). This pattern is consistent with pMP1 being acquired by the common

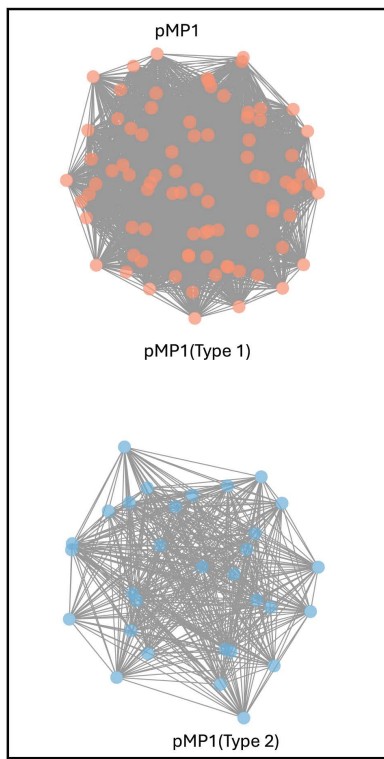
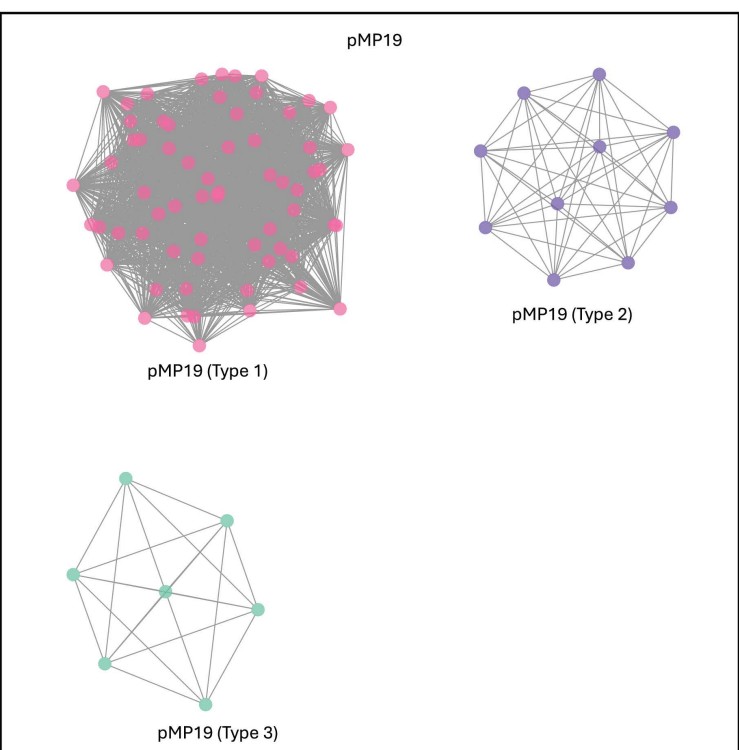

**Fig 7. K-mer clustering reveals genetic variation within pMP1 and pMP19 plasmids.** Networks were constructed using the k-mers shared between the strains. The solid colored circles (nodes) represent plasmids from a single isolate of *M. plutonius* and related nodes are connected by lines (edges). Networks with different colored nodes represent genetic subtypes of that plasmid.

ancestor followed by divergence and with evidence for horizontal movement of pMP1 plasmid among strains analyzed (Fig 7). A similar analysis of pMP19 revealed a different pattern. This plasmid is also common, being present in 92 out of 117 strains but it formed three subtypes. The pMP19 type 1 plasmid is present only in atypical strains. Conversely, the other two subtypes were identified in both typical and atypical strains, consistent with exchange of pMP19 subtypes types between the two groups of *M. plutonius*. We also identified pMP19 plasmid in 10 additional strains but the corresponding contigs had low read coverage (< 20) and/or assembled into multiple contigs. As a consequence, k-mer analysis failed to group these with the three subtypes.

During our analysis, we noted that a contig previously reported as plasmid pMP43 in strain B5 likely represents a bacteriophage rather than a plasmid. This sequence was not assembled as an independent molecule in any of the 91 strains sequenced in this study or in most publicly available genome sequences. The sequence of B5 is the only exception. In all other cases, we found the nearly complete sequence of pMP43 (bases 2,020–42,702) integrated into the chromosomes of all 14 typical strains from ST3 and ST7. This integrated region is 40.7 kb in length and has 99.99% homology to the full length of pMP43. The integrated pMP43 sequence is flanked by chromosomal segments and contains over 20 bacteriophage genes including those predicted to encode an integrase, terminase, lysozyme precursor, phase major and minor tail protein, capsid protein, phase portal protein, and other putative phage proteins. Likewise, a ~5 kb (bases 6,787–12, 604 of pMP43) or ~9 kb (bases 4,014 and 13, 207 of pMP43) fragment of pMP43 was found integrated in 15 atypical strains from ST12 and ST19 and shares 98% homology to pMP43. Given its likely prophage nature, we did not pursue further subtyping of pMP43.

## 3. Discussion

In the last decade, European foulbrood has become an increasing problem for US beekeepers [8]. In our study, rates of *M. plutonius* infection and EFB disease varied widely between years, operations, and even yards within an operation, and most operations had multiple strains (typical and atypical) with genetic variation (multiple genotypes) even at the level of a single honey bee colony. Additionally, many infected colonies were asymptomatic at the time of inspection suggesting that surveillance based on symptoms alone may be inadequate to track spread. Given the scale of migratory operations, some of which have thousands of honey bee colonies spread across dozens of yards around Michigan, sample size is likely not representative of the true infection rate within an operation, and further surveys are needed to characterize the true prevalence of this pathogen and disease in North America. Smaller stationary and hobby beekeeping operations also had high rates, but due to limited number of participants, this is likely not indicative of true rates within these operation types. However, even with small numbers of participants, hobby beekeepers had a very high rate of infection (7/16) which is surprising given the limited sampling (5 larvae) and the fact that most reported no signs of disease.

Pathogen prevalence reported here is consistent with previous studies in North America [7–9], and dramatically higher than rates reported in Europe [20]. We also report similar strain type (typical vs atypical) proportions as found by de León-Door et al. 2018 who reported the highest percentage of colonies screening positive for typical strains (71%), the lowest proportion screening positive for atypical strains only (3%), and 26% of positive results revealed both strains were present in the same colony. Slightly different proportions were found in Japan by Arai et al. (2014) which reported 40% typical strains, 6% atypical strains, and 54% of samples contained both strains [23]. Both typical and atypical strains of *M. plutonius* are commonly found together in infected colonies and as we show here, these coinfected colonies have significantly higher rates of clinical disease. However, whole genome analysis of isolates cultured from honey bee colonies that only screened positive for typical strains revealed that many coinfections are not detected by this screening and are likely under reported. Additionally, while the initial use of this duplex screening was shown to be sensitive and specific upon the initial publication (Arai et al. 2014), many bacteria in honey bees are still uncharacterized, making false positives possible.

We also found seasonal fluctuations in prevalence and disease rates consistent with a recent report by the USDA [8]. In that study, passive surveillance data reported a nearly identical rate of infection from Michigan submitted samples with

36% of the 170 samples they received screening positive for *M. plutonius* by microscopy [8,24]. They also observed a national increase in *M. plutonius* prevalence between 2021 and 2022 of approximately 10% and a seasonal peak in June consistent with our findings. However, the USDA passive surveillance is based on submissions from suspected diseased hives and may be an overestimation of actual pathogen prevalence, as healthy appearing colonies are likely not included. Inspection of colonies during travel for pollination also showed that EFB infection, disease, and consequently risk for transmission, occur early in the season, although infection and disease prevalence are significantly lower in February and March. It should be noted that our seasonal analysis was done with pooled data from multiple study designs, and samples taken outside of the peak prevalence period of June may skew results representing an overall underestimation of both pathogen and disease abundance over the course of the season. Additionally, many infected colonies were asymptomatic at the time of sampling, making control efforts based on symptoms alone difficult. While these asymptomatic colonies likely harbor lower hive-level pathogen loads [25], *M. plutonius* is known to survive for long periods in adult bees [26] which likely play a role in bee-mediated transmission by robbing or drift.

Whole genome SNP analysis revealed broad diversity, and very few related clusters were associated with a specific operation. Several genotypes (variants based on whole genome SNP analysis) were found to be shared between multiple operations suggesting a common source of infection. The most common genotype (Fig 6B – largest circle) was found in three migratory commercial beekeeping operations starting in 2022. This genotype was also found in a colony while in California for almond pollination and in other colonies while in Georgia prior to returning to Michigan. Given that this genotype was only found among migratory beekeeping operations and was not identified prior to 2022, it is possible that honey bee colonies from all three operations were in proximity at some point along the migratory route earlier in the year, prior to entering Michigan. However, drawing specific epidemiological links remains difficult, especially given that multiple genotypes can be found in a single honey bee colony, so the true genetic diversity within each operation remains unknown.

In addition to the broad diversity, we observed potential movement of a virulence associated plasmid pMP19, subtypes 2 and 3, between typical and atypical strains (Fig 5). Horizontal transfer of virulent plasmids between strains can have important implications, for example, transforming avirulent strains into virulent strains [27]. Nakamura et al. 2020 [15] showed a loss of virulence in CC3 strains in absence of pMP19 although CC12 strains remained virulent in absence of pMP19. It will be interesting to investigate if any of the pMP19 plasmid subtypes are more virulent and the outcomes of their interaction with strains in different clonal complexes and sequence types. Unfortunately plasmid presence or absence in the current study yields little insight for several reasons. First, cross-sectional surveillance that detected the pathogen may have missed associated disease due to the seasonal nature of EFB previously mentioned, so impact on hive health remains unclear. Second, given the previously mentioned abundance of co-infections, the strain sequenced may only represent one of many variants within the hive, so absence of plasmid in any sequence does not imply absence in the hive. Finally, pMP19 is occasionally lost after cultivation so plasmid loss through in vitro passage cannot be ruled out. Additional in vitro characterization of these plasmids may yield useful insights.

Sequencing data does provide some interesting insights into previously identified pMP43, notably sections integrated into the chromosome and phage associated genes suggest prophage origins, likely playing an important role in virulence of the strains harboring it and their interaction with the host. Prophages are known to influence virulence, metabolic pathways, stress response, and expression of host genes and are major drivers of genome evolution [28], representing a promising avenue for future EFB research.

Interestingly, sequence type 39 has only been identified in the United Kingdom by MLST and the variation at two loci, argE and gbpB have caused discrepancies with some placing this strain in CC3 and some placing it in CC13. Here we provide the first whole genome analysis of this strain which suggests it may be in a separate clonal complex CC39. We additionally confirm a close relationship between sequence type 3 and sequence type 7, shown in a similar core genome analysis.

While this study provides an important foundation for the genetic diversity of *M. plutonius* circulating in Michigan and the US, it is likely that the true diversity of this pathogen is under-represented. With PCR results revealing both typical and

atypical strains in nearly half of infected colonies, the single strain isolated for sequencing is likely not indicative of the genetic diversity within an infected hive. We additionally showed that duplex screening needs to be refined as detection of atypical strains seems to be less sensitive when typical strain coinfection is present. Expanded surveillance efforts and further characterization of pathogen diversity within infected hives will be critical for understanding both pathogenesis and transmission dynamics of this cryptic disease.

## 4. Methods

### 4.1 Selection of operations and yards for cross-sectional surveillance – 2021–2022 (Objective 1)

We enrolled beekeepers in a two-year, cross-sectional surveillance study. In 2021, we enrolled 10 operations: five migratory beekeeping operations (>500 hives) and five stationary beekeeping operations (between 50 and 500 hives). In 2022 the study included one additional migratory operation and four additional stationary operations. Additionally, in 2022 hobby beekeepers (<50 hives) were solicited through announcements to local beekeeping clubs and at the Michigan Beekeeping Conference to send in larval samples via mail. Within each stationary and commercial operation, we sampled colonies from multiple yards (locations where multiple honey bee colonies are managed as a group also referred to as an apiary) summarized in S1 Table. In 2021, yards and colonies within yards were selected at random and in 2022 convenience sampling was used to select yards as resources restricted random yard selection, but colonies in each yard were selected at random. In each yard, approximately 20 to 30% of the colonies were inspected for EFB and larval samples were collected for PCR screening. This was between 3 and 30 colonies depending on the size of the yard. Given a lack of baseline prevalence data, the number of hives sampled was based on resource availability. Colony selection for hobby operations was at the discretion of the beekeeper. Inspection and sampling occurred when the weather was optimal, from June 11th - August 17th, 2021, and from June 6th to August 31st in 2022. Sample size for objective 1 was limited by time and resource availability but was considered sufficient to establish baseline data needed to inform future studies (n = 855). To preserve anonymity, beekeeping operations are designated by an arbitrarily assigned alphabetic letter.

This study involved surveillance of honey bee colonies and analysis of bacterial pathogens. Beekeepers participated as research collaborators by providing access to their apiaries and assisting with sample collection. No human subjects research was conducted, and no personal data from beekeepers was collected or analyzed as part of this study.

### 4.2 Larval sample collection (Objective 1)

Larval sample selection was carried out following methods described by de León-Door et al., 2018, collecting 10 larvae, from across the brood nest when possible, into a 7 oz Whirl-Pak (Nasco, Fort Atkinson, WI, USA) containing 10 ml sterile PBS. If diseased larvae were found, early diseased but symptomatic larvae were selected as these are known to harbor higher loads of viable *M. plutonius* [29]. For hobby operation mail-in samples, beekeepers were provided with a sterile swab and sterile 7 oz Whirl-Pak and were instructed select hives at their discretion (symptomatic or non-symptomatic) and to use the swab to collect five larvae at their discretion (symptomatic if observed) into the Whirl-Pak and store in their freezer prior to mailing. Samples were mailed in pre-addressed envelopes using standard US postal service to Michigan State University. Once received, 5 ml PBS was added to each sample, and they were stored at −20º C until pathogen screening and cultivation could be performed as outlined below.

### 4.3 Colony inspection (Objective 1)

Of the 855 colonies included in 2021–2022 surveillance, 786 were inspected for visible signs of EFB disease (larvae that are discolored, visible trachea, twisted in the cell or melted) by evaluating each frame of brood in the brood nest. Hobby operations and two stationary operations from 2022 lacked hive health data (S1 Table). Inspections were carried out as described by Fowler et al. 2023 [4]. Briefly, the entire brood nest was examined for symptomatic larvae and scored based

on the total number of symptomatic larvae present, with colonies with fewer than 10 larvae showing signs considered healthy and colonies with 10 or more symptomatic larvae present considered clinically affected with EFB. All inspections were performed by a team of technicians trained in EFB diagnostics and colony health evaluation.

### 4.4 Duplex PCR screening (Objective 1)

A previously established duplex PCR [23] method was used to identify typical and atypical strains within the same larval sample by amplifying genes specific to each strain (S2 Table). Screening was performed using direct heat extraction as described in de León-Door et al., 2018 [7] with slight modifications listed here. Briefly, larvae were mechanically macerated in the Whirl-Pak and 200 μl of macerate was transferred to a sterile microcentrifuge tube. Larval macerate was then heated to 95ºC for 10 minutes, transferred to ice for 5 minutes and immediately subjected to direct PCR. Reactions were carried out with 2X GoTaq® Green Master Mix (Promega, Madison, WI, USA) and 0.4 μM forward and reverse primers in a 12 μl reaction with 2 μl of heat extracted larval homogenate as template. Amplification was carried out according to the program described by Arai et al. 2014 [23]. Amplicons were run on 1.2% agarose gel electrophoresis. Bands visualized at 424 base pairs (bp) were considered positive for atypical *M. plutonius* and bands at 187 bp were considered positive for typical *M. plutonius* [23]. Extracted DNA from *M. plutonius:* typical (ATCC 35311) and an atypical (regional isolate) (ST19) were used as positive controls and all runs included a template free negative control. Sensitivity was confirmed using previously analyzed samples 97.26% (95% CI: 90.45% − 99.67%) (S3 Table) [21].

### 4.5 Screening migratory colonies in Georgia and California (Objective 2)

Colonies were screened in Georgia in April while in a holding yard (2022) and in California in early February during almond pollination (2023). In April 2022 in Georgia, two holding yards belonging to two Michigan based migratory beekeepers (C and K) were selected for screening. Within each yard, 99 colonies for operation C and 29 colonies for operation K (n = 128 total) were selected by convenience sampling. Colony inspection and sample handling were as previously described in section 4.1 to 4.4. During February 2023 in California, we sampled larvae from 144 colonies, from five migratory Michigan-based operations (A, C, D, E and F) and from an additional 17 operations from different states (S4 Table). Yards and colonies were chosen by convenience sampling and five colonies per yard were included. Presence of clinical disease was determined based on previously described visual signs. A sterile culture swab was used to macerate five larvae within their cells to expose midgut contents. Swabs were kept sealed at ambient temperature or on ice if available and shipped to Michigan State University for processing. Prior to screening, swabs were submerged and agitated in 2 ml of PBS to dissolve larval macerate, which was then used for PCR duplex screening as previously described. A lower volume of PBS was used (less than 1 ml per larvae) than in section 4.2 to account for the reduced mass picked up by the swab.

### 4.6 Culture and isolation of *M. plutonius* (Objective 3)

All larval samples which screened positive for *M. plutonius* by the previously mentioned duplex PCR (n = 308) were cultivated and purified to prepare for sequencing by streaking 1 μl of larval macerate on freshly prepared M110 agar media [30]. Plates were incubated in an anaerobic chamber under 10% $CO_2$, 5% $H_2$, 85% $N_2$ at 37º C. Plates were checked daily (until day 11) for round smooth white colonies approximately 1 mm in diameter, consistent with *M. plutonius*. Samples showing overgrowth consistent with *Paenibacillus alvei* (Ex. White) [24] were re-cultured on M110 plates supplemented with 3 μl/ml nalidixic acid [30]. Colonies consistent with *M. plutonius* [30] were selected with a sterile toothpick, transferred to 5 ml freshly prepared KSBHI broth (BHI supplemented with 0.15M $KH_2PO_4$ and 1% soluble starch) and incubated anaerobically at 37º C, agitating daily for 3–5 days. Plates streaked with regionally isolated atypical control and typical control (ATCC 35311) were included in each round of incubation. Bacterial DNA was extracted (n = 171) and bacteria identity was confirmed by 16S PCR as described by Fowler et al. 2023 [4].

## 4.7 Isolate selection and whole genome sequencing (Objective 3)

A subset of *Melissococcus plutonius* isolates (n = 96) were selected for whole genome sequencing. This included isolates from hives sampled in this study (2022–2023), as well as isolates from previously published projects collected in 2019 [21] and 2020 [4]. We selected isolates to represent the broadest temporal and geographic range, selecting 1–2 from each operation, across multiple yards and multiple years, when possible, to get a broad picture of pathogen diversity. Extracted DNA was submitted to Michigan State University Genomics Core for library preparation and sequencing. Libraries were prepared using Roche Kapa HyperPrep DNA Library Preparation Kit (Roche Sequencing Solutions, Pleasanton, CA, USA) and pooled samples were sequenced (150 base pair paired-end) using NovaSeq 6000 (Illumina, San Diego, CA, USA). Raw reads were deposited on NCBI Sequence Read Archive (SRA) under bioproject PRJNA1195964. Accession numbers for all isolates are outlined in supplementary file Milbrath-Mplutonius-genomes-this-study.

## 4.8 Bioinformatics and population structure (Objective 3)

### 4.8.1 Short read processing, assembly, and genome annotation.
To process and analyze sequencing data, an automated workflow was implemented using the nf-core/pathogensurveillance pipeline (https://nf-co.re/pathogensurveillance/dev; Foster et al., manuscript in preparation). Key software, with default settings unless indicated, used in the workflow included the following. Fastp version 0.23.4 [31] was used to trim adapter sequences from reads and Spades version 3.15.5 [32] was used to assemble reads. Contigs with fewer than 500 base pairs were filtered out and Bakta version 1.9.4 [33] was used to annotate genome sequences. QUAST version 5.2.0 [34] was used to assess assembly quality. For variant calling, raw reads were aligned to the references with BWA mem (BWA version 0.7.17-r1188) [35] and single nucleotide polymorphisms (SNPs) were called with graphtyper version 2.7.2 [36]. For 26 genome assemblies retrieved from NCBI, ART version 2.5.8 [37] was used to simulate 150 bp paired-end short reads and a 50 x coverage. Genome sequences of strains ATCC 35311 and DAT 561 (GCF_003966875.1) were used as references for typical and atypical strains, respectively. To visualize relationships between genotypes, a minimum spanning network was constructed with poppr version 2.9.6 [38]. Strains were clustered into genotypes based on a threshold of ≤ 15 SNP differences. The 15 SNP threshold was determined based on genome size and read quality [39].

### 4.8.2 Phylogenomic analysis and sequence typing.
A core-genome tree consisting of both typical and atypical strains and nine Enterococcous strains was constructed. The program, PIRATE, version 1.5.0 [40], was used to identify orthologs among genome sequences. Sequences of 549 single copy orthologs were extracted with a custom script and MAFFT version 7.526 [41] was used to generate a multiple sequence alignment. A phylogenetic tree was obtained with IQ-TREE version 2.1.4-beta [42], rooted to *Enterococcous casseliflavus* NCTC12362 and visualized with ggtree 3.1.0 [43]. A heatmap was added to the tree, using function gheatmap in ggtree. To infer sequence types of sequenced strains, we queried the assembled genomes to PubMLST database [44].

### 4.8.3 Plasmid analysis.
We identified plasmid sequences by using MUMmer to map assembled contigs from *M. plutonius* assemblies to reference plasmids pMP1 and pMP19 [45]. Contigs were considered plasmid-derived when they satisfied three criteria: (1) collectively covered the majority of the reference plasmid sequence, (2) had a minimum mapping length of 500 bp, and (3) mapped uniquely to a single region of the reference. Identified plasmid contigs were extracted using SAMtools (version 1.18) [46] and consolidated into a single fasta file per strain per plasmid type. We then used PopPunk 2.6.5 to cluster pMP1 and pMP19 sequences into genetic groups [47] and Cytoscape 3.10.2 [48] to visualize plasmid relationships.

## 4.9 Statistical analysis

All figures were generated and all statistical analyses were performed in R version 4.3.0 [49,50]. Differences in disease and infection rates among groups were compared using a chi-squared test and assumptions for independence and

expected frequencies were met. Differences with $p \leq 0.05$ were considered statistically significant. Figures were edited in Adobe Illustrator version 24.1.2 (Adobe Inc. San Jose, CA, USA).

## Supporting information

**S1 Table. Number of colonies and number of PCR positive colonies sampled per yard and operation for surveillance study (n = 855).** Yards were numbered in the order they were sampled and differed for each operation between years.
(DOCX)

**S2 Table. PCR primers used in this study.**
(DOCX)

**S3 Table. Duplex screening sensitivity confirmation – Screening was performed on previously analyzed samples and compared to results from qPCR performed on the same samples at USDA as part of another validation study (Milbrath et al. 2021).**
(DOCX)

**S4 Table. Operations sampled during almond pollination in California.** Home state represents the primary state where the beekeeping operation resides.
(DOCX)

**S1 Fig. Example gel from direct duplex screening.** Supplement.zip – Bioinformatics metadata for sequences used in this study.
(TIF)

## Acknowledgments

Thanks to Heather Chapman, Ana Heck, Robyn Hawley, Dan Wyns, Cade Houston, Ben Sallmann and all the others that contributed to field data collection as well as the beekeepers who agreed to participate in this study. Special thanks to Azam Sher and HC for assistance with lab work, Drs. Zach Foster, Camilo Parada-Rojas, and Martha Sudermann for their advice with data analyses, and to the Mansfield lab for providing the laboratory space. We also thank the Department of Botany and Plant Pathology at Oregon State University for its continual support of the high-performance computing infrastructure.

## Author contributions

**Conceptualization:** Peter Fowler, Jeff H. Chang, Meghan O. Milbrath.

**Data curation:** Peter Fowler, Upasana Dhakal.

**Formal analysis:** Peter Fowler, Upasana Dhakal, Jeff H. Chang.

**Funding acquisition:** Peter Fowler, Jeff H. Chang, Meghan O. Milbrath.

**Investigation:** Peter Fowler, Upasana Dhakal, Jeff H. Chang, Meghan O. Milbrath.

**Methodology:** Peter Fowler, Upasana Dhakal, Jeff H. Chang.

**Project administration:** Peter Fowler, Meghan O. Milbrath.

**Resources:** Peter Fowler, Meghan O. Milbrath.

**Software:** Jeff H. Chang.

**Supervision:** Peter Fowler, Jeff H. Chang, Meghan O. Milbrath.

**Validation:** Peter Fowler, Upasana Dhakal, Meghan O. Milbrath.

**Visualization:** Peter Fowler, Upasana Dhakal.

**Writing – original draft:** Peter Fowler, Upasana Dhakal, Jeff H. Chang.

**Writing – review & editing:** Peter Fowler, Upasana Dhakal, Jeff H. Chang, Meghan O. Milbrath.

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
