## [Decision Letter · Decision Letter 0]

26 Jun 2025

*Melissococcus plutonius*

Dear Dr. Fowler,

Thank you for submitting your manuscript to PLOS ONE. After careful consideration, we feel that it has merit but does not fully meet PLOS ONE’s publication criteria as it currently stands. Therefore, we invite you to submit a revised version of the manuscript that addresses the points raised during the review process.

We look forward to receiving your revised manuscript.

Kind regards,

Kai Wang

Academic Editor

PLOS ONE

Journal Requirements:

5. We notice that your supplementary figures are uploaded with the file type 'Figure'. Please amend the file type to 'Supporting Information'. Please ensure that each Supporting Information file has a legend listed in the manuscript after the references list.

6. Please include a copy of Table S4 which you refer to in your text on page 25.

Reviewers' comments:

Reviewer's Responses to Questions

**Comments to the Author**

1. Is the manuscript technically sound, and do the data support the conclusions?

Reviewer #1: Yes

Reviewer #2: Yes

Reviewer #3: Yes

2. Has the statistical analysis been performed appropriately and rigorously?

Reviewer #1: Yes

Reviewer #2: Yes

Reviewer #3: Yes

3. Have the authors made all data underlying the findings in their manuscript fully available?

Reviewer #1: Yes

Reviewer #2: Yes

Reviewer #3: Yes

4. Is the manuscript presented in an intelligible fashion and written in standard English?

Reviewer #1: Yes

Reviewer #2: Yes

Reviewer #3: Yes

Reviewer #1: Your manuscript presents highly informative and valuable data on the asymptomatic and symptomatic prevalence of M. plutonius in US honey bee colonies. The revealed high genetic diversity offers crucial insights for future mitigation strategies and prevalence studies. I've included a few comments and suggestions in the attached PDF to enhance the clarity of certain methodological aspects and the discussion section for the reader. This manuscript deserves publication after minor revisions.

Reviewer #2: The authors combined surveillance and molecular epidemiology to investigate prevalence, diversity, and transmission dynamics of Melissococcus plutonius in US honey bee colonies. The results are promising and gave epidemiological insight into EFB disease in US.

The manuscript could benefit from further context on the role of adult bees as potential asymptomatic carriers of M. plutonius and the implications for disease transmission. Could the authors elaborate on this aspect and discuss its potential impact on colony-level epidemiology? Additionally, given that co-infections with typical and atypical M. plutonius strains are commonly reported in EFB, did the authors observe any association between virulence factors—particularly the plasmids pMP1 and pMP19—and the presence of both strain types within a single colony? Were pMP1 and pMP19 detected concurrently in co-infected colonies? Exploring this might enhance understanding of plasmid-mediated virulence and its distribution across strain types.

Reviewer #3: L108-109: formatting error

L120/122: Oregon-based or Oregon based

L138/140: Please check italic species names throughout manuscript

L188: italic species names, please check throughout document

L194: punctuation missing.

L195ff: Is there a new paragraph or does it all belong to figure caption? Please cite/link your metadata file with the genotypes numbers.

L204: Please check throughout manuscript: Fig-6, or Fig. 6, or Fig 6 ?

L229: do you refer to your own results or other’s not cited here?

L237: subtypes types

L272:Are there more/additional supporting publications for North America than just the Mexican study?

L282: under-reported

L287: Please cite the reference here already (8?) as you refer to it in next sentence, otherwise bit confusing.

L356: space between 10 ml and mL. Units seem quite inconsistently used throughout the manuscript. Please check again, i.e. ml, mL, uL, ul, and spaces between numbers and units

L360: Is there an overgrowth with secondary bacteria expected if samples are not shipped on ice/cooled – in comparison to your own collected samples?

L394: “previously described […] study” not citied

L241: neither figure caption nor graph is very explanatory. Are the single colored dots in each network corresponding to the 96 isolates?

L402: did you mean ‘section 4.2’ instead of 2.2?

L409: was P. alvei the only species overgrowing your plates?

L412: please check spaces between numbers and SI units

L416: Incomplete sentence, please check.

L428: file in suppl. Material is called differently: Mplutonius-public-assemblies-NCBI-accessions?

L432: odd sentence structure “we used an automated workflow was implemented using....”

L442: which reference did you use (accession number), bcs you have two DAT561 in your supplement file? The 2018 assembly?

L444: Did you find a reference stating less than15 SNPs difference are defined for clustering bacterial strains or Mplutonius?

L496: Is there no year and publisher available?

From L479: please check again: some Journal names were abbreviated but many are not (520, 526, 531, 534…).

L602, 605: Font size is bigger than others in this section

Appendix:

Figure 2 and 4: major tick marks missing. Maybe include (faded) horizontal gridlines consistent for all graphs. It’s untypical to use “0%, 25%, 50% “ for scientific graphs as there is the word “Percent” in the axis title included anyways

Figure 6: I had some trouble to understand where your referred genotypes (f.e. 14, 33, and 26) are to find in this graph. In the Results section you refer to Fig 6 when explaining that populations in certain operations consist of genotype x, y, z but this is not the information I can retrieve from this graph. Do you maybe refer to Suppl. File Mplutonius-genotypes-plasmids-metadata_table2?

**Do you want your identity to be public for this peer review?** For information about this choice, including consent withdrawal, please see our Privacy Policy

Reviewer #1: **Yes: ** Marco Pietropaoli

Reviewer #2: No

Reviewer #3: No

---

## [Author Response · Author response to Decision Letter 1]

15 Aug 2025

Reviewer #1: Your manuscript presents highly informative and valuable data on the asymptomatic and symptomatic prevalence of M. plutonius in US honey bee colonies. The revealed high genetic diversity offers crucial insights for future mitigation strategies and prevalence studies. I've included a few comments and suggestions in the attached PDF to enhance the clarity of certain methodological aspects and the discussion section for the reader. This manuscript deserves publication after minor revisions.

Reviewer #2: The authors combined surveillance and molecular epidemiology to investigate prevalence, diversity, and transmission dynamics of Melissococcus plutonius in US honey bee colonies. The results are promising and gave epidemiological insight into EFB disease in US.

The manuscript could benefit from further context on the role of adult bees as potential asymptomatic carriers of M. plutonius and the implications for disease transmission. Could the authors elaborate on this aspect and discuss its potential impact on colony-level epidemiology?

We’ve added the following (Line 323):

Additionally, many infected colonies were asymptomatic at the time of sampling, making control efforts based on symptoms alone difficult. While these asymptomatic colonies likely harbor lower hive-level pathogen loads [25], M. plutonius is known to survive for long periods in adult bees [26] which likely play a role in bee-mediated transmission by robbing or drift.

Additionally, given that co-infections with typical and atypical M. plutonius strains are commonly reported in EFB, did the authors observe any association between virulence factors—particularly the plasmids pMP1 and pMP19—and the presence of both strain types within a single colony? Were pMP1 and pMP19 detected concurrently in co-infected colonies? Exploring this might enhance understanding of plasmid-mediated virulence and its distribution across strain types.

We’ve added the following to clarify what can be drawn from whole genome sequencing data with respect to plasmid presence/absence (Line 347).

Unfortunately plasmid presence or absence in the current study yields little insight for several reasons. First, cross-sectional surveillance that detected the pathogen may have missed associated disease due to the seasonal nature of EFB previously mentioned, so impact on hive health remains unclear. Second, given the previously mentioned abundance of co-infections, the strain sequenced may only represent one of many variants within the hive, so absence of plasmid in any sequence does not imply absence in the hive. Finally, pMP19 is occasionally lost after cultivation so plasmid loss through in vitro passage cannot be ruled out. However, sequencing data does provide some interesting insights into previously identified pMP43, notably sections integrated into the chromosome and phage associated genes suggest prophage origins, likely playing an important role in virulence of the strains harboring it and their interaction with the bee larvae. Prophages are known to influence virulence, metabolic pathways, stress response, and expression of host genes and are major drivers of genome evolution [31], representing a promising avenue for future EFB research.

Reviewer1/2 specific comments in PDF are summarized here in the order they were given.

Line 22: Please specify if in this case the 50% of infection is clinical, subclinical or both

We’ve changed from:

Rates of infection and disease were found to be seasonal, with prevalence peaking in June when over half the colonies screened were infected.

To:

(Line 25) Infection and disease were found to be seasonal, with prevalence of both peaking in June when over half the colonies were infected, and over 20% had clinical EFB.

Line 66: Please specify that not only strains collected with the surveillance study were analysed but also those from previously published projects collected in 2019 and 2020.

We’ve changed from:

Finally, whole genome sequences of circulating strains of M. plutonius were analyzed to both infer phylogenetic relationships, and to provide insights into pathogen diversity, transmission, and persistence (objective 3).

To:

(Line 72) Finally, a subset of M. plutonius isolates from the current surveillance study, and isolates collected from previous projects dating back to 2019 underwent whole genome analysis to both infer phylogenetic relationships, and to provide insights into pathogen diversity, transmission, and persistence (objective 3).

Line 94: I suggest to specify "by duplex PCR"

Line 94: I suggest to move it to M&M

We’ve changed from:

Of the 855 colonies screened from 2021 to2022, 787 were inspected for clinical disease, 284 of these screened positive for M. plutonius, and 28% of these positive colonies (n=80) had signs of EFB (more than 10 symptomatic larvae) at the time of sampling.

To:

(Line 102) Of the 855 colonies screened from 2021 to2022, 787 were inspected for clinical disease, 284 of these screened positive for M. plutonius by duplex PCR, and 28% of these positive colonies (n=80) had signs of EFB at the time of sampling.

Line 108: move upper line

We’ve made this modification.

Line 248: remove

We’ve removed the extra comma, thanks for catching that.

Line 329: Further suggestions for future studies could be added at the end of the manuscript including improvements in the methodology/data collection (e.g. collection of samples/identification of the disease on field by beekeepers - line 271) and lab analysis (e.g. how to avoid/reduce the duplex screening false positives - line 284) used to assess the presence of M. plutonius.

We’ve made the following modification from:

Additional efforts to characterize this diversity will be critical for understanding both pathogenesis and transmission dynamics of this cryptic disease.

To:

(Line 372) We additionally showed that duplex screening needs to be refined as detection of atypical strains seems to be less sensitive when typical strain coinfection is present. Expanded surveillance efforts and further characterization of pathogen diversity within infected hives will be critical for understanding both pathogenesis and transmission dynamics of this cryptic disease.

Line 337: Please specify how hobby beekeepers were involved (mailinglists, social media, etc.)

We’ve changed from:

In 2022 the study included one additional migratory operation and four additional stationary operations, as well as hobby beekeepers (<50 hives), who send in samples via mail.

To:

(Line 383) In 2022 the study included one additional migratory operation and four additional stationary operations. Additionally, in 2022 hobby beekeepers (<50 hives) were solicited through announcements to local beekeeping clubs and at the Michigan Beekeeping Conference to send in larval samples via mail.

Line 341: Please justify this methodological shift

We’ve changed from:

In 2021, yards and colonies within yards were selected at random and in 2022 convenience sampling was used to select yards, but colonies in each yard were selected at random.

To:

(Line 388) In 2021, yards and colonies within yards were selected at random and in 2022 convenience sampling was used to select yards as resources restricted random yard selection, but colonies in each yard were selected at random.

Line 342: The number of colonies inspected was determined considering an expected prevalence? The range size of the apiaries had a minimum and maximum?

We’ve changed the text from:

In each yard, approximately 20 to 30% of the colonies were inspected for EFB and larval samples were collected for PCR screening. This was between 3 and 30 colonies depending on the size of the yard. Colony selection for hobby operations was at the discretion of the beekeeper.

To:

(Line 390) In each yard, approximately 20 to 30% of the colonies were inspected for EFB and larval samples were collected for PCR screening. This was between 3 and 30 colonies depending on the size of the yard. Given a lack of baseline prevalence data, the number of hives sampled was based on resource availability.

Line 347: add “to”

We’ve made the requested change.

Line 359: What were the criteria that beekeepers used to select the colonies from which to collect the samples (e.g., with clinical symptoms)? Did selected larvae have to show clinical symptoms?

We’ve made the following change from:

For hobby operation mail-in samples, beekeepers were provided a sterile swab and sterile 7oz Whirl Pak and were instructed to use the swab to collect five larvae into the Whirl-Pak and store in their freezer prior to mailing.

To:

(Line 409) For hobby operation mail-in samples, beekeepers were provided with a sterile swab and sterile 7oz Whirl-Pak and were instructed select hives at their discretion (symptomatic or non-symptomatic) and to use the swab to collect five larvae at their discretion (symptomatic if observed) into the Whirl-Pak and store in their freezer prior to mailing.

Line 368: Please add the detail (according to Fowler): "colonies with 10 cells or fewer showing symptoms were considered healthy, while colonies with more than 10 cells affected were considered diseased"

We’ve changed the text to clarify as suggested from:

Inspections were carried out as described by Fowler et al. 2023 (4).

To:

(Line 421) Inspections were carried out as described by Fowler et al. 2023 [4]. Briefly, the entire brood nest was examined for symptomatic larvae and scored based on the total number of symptomatic larvae present, with colonies with fewer than 10 larvae showing signs considered healthy and colonies with 10 or more symptomatic larvae present considered clinically affected with EFB.

Line 416: Incomplete sentence.

We’ve removed this.

Line 448: Enterococcus. Please check also in the rest of the manuscript

We’ve removed removed the italic and checked throughout the manuscript.

Line 449: uppercase?

We’ve changed pirate to PIRATE and made corrections to MAFFT and IQ-TREE.

Reviewer #3: L108-109: formatting error

Thanks for catching this, we’ve made the change.

L120/122: Oregon-based or Oregon based

We’ve removed the hyphen.

L138/140: Please check italic species names throughout manuscript

We’ve double checked italic species names throughout the manuscript.

L188: italic species names, please check throughout document

We’ve double checked italic species names throughout the manuscript.

L194: punctuation missing.

We’ve added a period.

L195ff: Is there a new paragraph or does it all belong to figure caption? Please cite/link your metadata file with the genotypes numbers.

We’ve updated the graphic to highlight the genotype numbers referenced in the text.

L204: Please check throughout manuscript: Fig-6, or Fig. 6, or Fig 6 ?

Thank you, we’ve normalized these naming conventions.

L229: do you refer to your own results or other’s not cited here?

We’ve added a citation here to a comparative genomics analysis.

L237: subtypes types

We’ve removed “types”

L272:Are there more/additional supporting publications for North America than just the Mexican study?

Yes, we’ve added a study from Oregon and passive surveillance study previously mentioned in the introduction.

L282: under-reported

We’ve added the requested hyphen.

L287: Please cite the reference here already (8?) as you refer to it in next sentence, otherwise bit confusing.

We’ve added the citation.

L356: space between 10 ml and mL. Units seem quite inconsistently used throughout the manuscript. Please check again, i.e. ml, mL, uL, ul, and spaces between numbers and units

We’ve gone through the manuscript and standardized these units.

L360: Is there an overgrowth with secondary bacteria expected if samples are not shipped on ice/cooled – in comparison to your own collected samples?

We’ve outlined in the culture section how overgrowth was handled to obtain isolates (addition of nalidixic acid).

L394: “previously described […] study” not citied

We’ve changed to clarify sections previously referenced by changing from:

Colony inspection and sample handling were as previously described for the 2021-2022 Michigan surveillance study.

To:

(Line 450) Colony inspection and sample handling were as previously described in section 4.1 to 4.4.

L241: neither figure caption nor graph is very explanatory. Are the single colored dots in each network corresponding to the 96 isolates?

We’ve changed the caption for clarity from:

Networks were constructed using the k-mers shared between the strains. The solid colored circles are the nodes representing plasmids and nodes are connected by lines (edges). Networks with different colored nodes represent genetic subtypes of a plasmid.

To:

Networks were constructed using the k-mers shared between the strains. The solid colored circles (nodes) represent plasmids from a single isolate of M. plutonius and related nodes are connected by lines (edges). Networks with different colored nodes represent genetic subtypes of that plasmid.

L402: did you mean ‘section 4.2’ instead of 2.2?

Yes, we’ve made the correction.

L409: was P. alvei the only species overgrowing your plates?

Yes, other species form isolated colonies and rarely overgrow plates.

L412: please check spaces between numbers and SI units

We’ve added spaces throughout.

L416: Incomplete sentence, please check.

We’ve removed this sentence.

L428: file in suppl. Material is called differently: Mplutonius-public-assemblies-NCBI-accessions?

We’ve corrected the referenced file with deposited strains. Milbrath-Mplutonius-genomes-this-study

L432: odd sentence structure “we used an automated workflow was implemented using....”

We’ve made the following change from:

To process and analyze sequencing data, we used an automated workflow was implemented using the nf-core/pathogensurveillance pipeline (https://nf-co.re/pathogensurveillance/dev; Foster et al., manuscript in preparation).

To:

(Line 491) To process and analyze sequencing data, an automated workflow was implemented using the nf-core/pathogensurveillance pipeline (https://nf-co.re/pathogensurveillance/dev; Foster et al., manuscript in preparation).

L442: which reference did you use (accession number), bcs you have two DAT561 in your supplement file? The 2018 assembly?

Yes, it was the 2018 assembly and is listed in our core genome metadata file, but we’ve clarified by adding (GCF_003966875.1) here (Line 503).

L444: Did you find a reference stating less than 15 SNPs difference are defined for clustering bacterial strains or Mplutonius?

We’ve added some rational and a reference to previous research (Line 506).

The 15 SNP threshold was determined based on genome size and read quality (37).

L496: Is there no year and publisher available?

We’ve updated this reference.

From L479: please check again: some Journal names were abbreviated but many are not (520, 526, 531, 534…).

We’ve normalized journal name conventions.

L602, 605: Font size is bigger than others in this section

We’ve made the entire bibliography size 12 font.

Appendix:

Figure 2 and 4: major tick marks missing. Maybe include (faded) horizontal gridlines consistent for all graphs. It’s untypical to use “0%, 25%, 50% “ for scientific graphs as there is the word “Percent” in the axis title included anyways

We’ve added horizontal gridlines and removed the percentage for consistency.

Figure 6: I had some trouble to understand where your referred genotypes (f.e. 14, 33, and 26) are to find in this graph. In the Results section you refer to Fig 6 when explaining that populations in certain operations consist of genotype x, y, z but this is not the information I can retrieve from this graph. Do you maybe refer to Suppl. File Mplutonius-genotypes-plasmids-metadata_table2?

We’ve updated this figure to highlight the referenced genotypes.

---

## [Editor Report · Decision Letter 1]

25 Aug 2025

Everything, everywhere, all at once - Surveillance and molecular epidemiology reveal Melissococcus plutonius is endemic among Michigan, US beekeeping operations

PONE-D-25-28058R1

Dear Dr. Fowler,

We’re pleased to inform you that your manuscript has been judged scientifically suitable for publication and will be formally accepted for publication once it meets all outstanding technical requirements.

Kind regards,

Kai Wang

Academic Editor

PLOS ONE

Additional Editor Comments (optional):

A great work, congratulations to you and your team!

Regards,

Kai WANG, Editor

kaiwang628@gmail.com
---

## [Editor Report · Acceptance letter]

PONE-D-25-28058R1

PLOS ONE

Dear Dr. Fowler,

I'm pleased to inform you that your manuscript has been deemed suitable for publication in PLOS ONE. Congratulations! Your manuscript is now being handed over to our production team.

Kind regards,

on behalf of

Dr. Kai Wang

Academic Editor

PLOS ONE